# Experimental Approaches in Delineating mTOR Signaling

**DOI:** 10.3390/genes11070738

**Published:** 2020-07-02

**Authors:** Jiayi Qian, Siyuan Su, Pengda Liu

**Affiliations:** 1Lineberger Comprehensive Cancer Center, The University of North Carolina at Chapel Hill, Chapel Hill, NC 27599, USA; qjysusan@126.com (J.Q.); susy@email.unc.edu (S.S.); 2Department of Biochemistry and Biophysics, The University of North Carolina at Chapel Hill, Chapel Hill, NC 27599, USA

**Keywords:** mTOR, experimental approach, biochemical approach, genetic approach, hypothesis-driven, immunofluorescence, protein motif search, bioinformatic approach

## Abstract

The mTOR signaling controls essential biological functions including proliferation, growth, metabolism, autophagy, ageing, and others. Hyperactivation of mTOR signaling leads to a plethora of human disorders; thus, mTOR is an attractive drug target. The discovery of mTOR signaling started from isolation of rapamycin in 1975 and cloning of TOR genes in 1993. In the past 27 years, numerous research groups have contributed significantly to advancing our understanding of mTOR signaling and mTOR biology. Notably, a variety of experimental approaches have been employed in these studies to identify key mTOR pathway members that shape up the mTOR signaling we know today. Technique development drives mTOR research, while canonical biochemical and yeast genetics lay the foundation for mTOR studies. Here in this review, we summarize major experimental approaches used in the past in delineating mTOR signaling, including biochemical immunoprecipitation approaches, genetic approaches, immunofluorescence microscopic approaches, hypothesis-driven studies, protein sequence or motif search driven approaches, and bioinformatic approaches. We hope that revisiting these distinct types of experimental approaches will provide a blueprint for major techniques driving mTOR research. More importantly, we hope that thinking and reasonings behind these experimental designs will inspire future mTOR research as well as studies of other protein kinases beyond mTOR.

## 1. Introduction

As one of the most abundant gene families in human, protein kinases tightly control cell signaling and cell function through protein phosphorylation. There are 634 protein kinases encoded in human genome to date, and ~85% kinases are observed dysregulated in human disorders [1]. However, only 49 small molecule kinase inhibitors have been approved for treating human diseases [2], urging for more in-depth investigations on the rest 77% possibly druggable kinome. Among all protein kinases, mTOR (mechanistic target of rapamycin, previously known as mammalian target of rapamycin) has drawn extensive attention. This is due to its indispensable roles in regulating key cell function such as proliferation, metabolism, autophagy, ageing, and others, and its ability to sense environmental changes including nutrients and growth factors to adjust cell physiology. Dysregulation of mTOR signaling is observed in solid tumors, epilepsy, obesity, and diabetes. Hyperactivation of mTOR signaling is believed to drive tumorigenesis supported by both clinical data and rodent models [3,4,5,6]. From cloning of TOR genes by the Hall group in 1993 [7] to currently advanced understanding of mTOR biology, multiple experimental techniques/approaches have been applied to identify key mTOR pathway components to delineate mTOR signaling. In this review, we focus on major and distinct experimental approaches used in early work that led to identification and characterization of key mTOR signaling pathway members. The journey of mTOR research starts from isolation of rapamycin as an immune suppressant and mTORC1 inhibitor, and is greatly expanded by traditional genetic approaches and biochemical protein purification-mediated studies. Due to advances of technology, more recent studies tend to rely on large-scale nonbiased screening approaches for discoveries. In addition, immunofluorescence, bioinformatic, and hypothesis-driven approaches or a combination of different techniques are indispensable to advance our understanding of mTOR biology, regulation, and function. We hope that these experimental approaches and inspiring logics behind in this review will benefit future research related to mTOR and other protein kinases.

## 2. Overview of mTOR Signaling

mTOR is present in two structurally distinctive mTOR protein complexes, namely, mTORC1 and mTORC2. Both complexes share common subunits mTOR and GβL, whereas Raptor depicts mTORC1 and Rictor/Sin1 uniquely earmark mTORC2 [8]. Activation of mTORC2 is largely induced by growth signaling, and mTORC1 responds to both growth stimulations from mTORC2/Akt/TSC2/Rheb signaling and availability of amino acids through Rag signaling [5,6]. Given to its pivotal roles in controlling essential cellular processes and physiological functions, activation of mTORC1 and mTORC2 is tightly controlled in both tempo and spatial manners [4]. This is achieved by a hierarchy of protein cascade-mediated mTORC1 activity control mechanisms on lysosome (detailed in later sections in this review) or by a variety of post-translational modifications (such as phosphorylation [9] and ubiquitination [10]) to acutely respond to cellular and environmental cues. Due to the large entity of all mTOR-related research articles published in the past and limited space in this review, we will only focus on discoveries that lay the foundation to shape up the mTOR signaling pathway, although important and disease-related, transcriptional, translational, and post-translational regulations of the mTOR signaling will not be discussed in this review. We also apologize to colleagues whose important and fabulous work could not be mentioned in this review due to page limits.

## 3. Identification of *TOR* Genes by Yeast Genetics and Biochemical Purification Approaches as Rapamycin Effectors

Distinct from kinases such as p44MAPK [11], whose discovery was initiated by cloning of the kinase gene, the journey of mTOR signaling starts from isolation of an antifungal compound rapamycin from soil on Easter Island in 1975 by Suren N. Sehgal [12]. Rapamycin demonstrated an activity against *Candida albicans* but not Gram-positive and -negative bacteria. Although earlier work revealed that rapamycin was immunosuppressive [13] and impeded cancer growth [14], the underlying mechanisms for rapamycin function was halted until rapamycin was broadly available to researchers.

Rapamycin was observed to arrest eukaryotic cells at the G1 phase in cell cycle. A genetic search for mutations that confer to rapamycin resistance in budding yeast led to identification of *TOR1* and *TOR2* genes as rapamycin effectors [15]. Specifically, spontaneous independent mutants resistant to rapamycin (0.1 μg/mL) were isolated from a haploid derivative of yeast strain JK9-3d. Genetic crosses between rapamycin-resistant and -sensitive parental strains demonstrated that the majority of the mutations (15 of 18) were fully recessive (diploids were rapamycin sensitive). Two mutations, labeled as *tor1-1* and *tor2-1* for target of rapamycin, were recessive at 24 °C, while were partially dominant at 30 °C or 37 °C. The one fully dominant mutant, *tor1-2*, was isolated from the strain JK9-3d. Examination of crosses between the *tor1-1* and *tor2-1* mutants initially suggested that these two mutations were nonallelic because they failed to complement; a *tor1-1*tor2-1* diploid was resistant to rapamycin (10 μg/mL) under conditions where the two mutations were recessive (24 °C). However, the two mutations segregated independently in meiosis [15], therefore they were defined as two independent genes (*TOR1* and *TOR2*). Two years later, an independent study isolating dominant rapamycin-resistant mutants using yeast genetics also pinpointed *DRR1/DRR2* (which are *TOR*s) as targets mediating the toxicity activity of rapamycin in yeast [16]. 

Subsequently, the connection between rapamycin and TORs was revealed by both biochemical and genetic approaches. Biochemically, the cellular binding partner(s) for rapamycin was identified as FKBP12, which is shared by FK506 [17], a related immunosuppressive compound that also blocks helper T cell activation. FK506 and rapamycin bind to the same site of FKBP12 but with different downstream targets [18]. Taking advantage of this knowledge, FKBP12-rapamycin complexes were used as the bait to fish for binding partners, and mTOR was found as the hit. Specifically, Sabatini et al. relied on ^32^P-labeled FKBP12 proteins with or without rapamycin to pulldown PC12 pheochromocytoma cell lysates, followed by DSS (disuccinimidyl suberate)-mediated cross-linking. This led to identification of a 260 KD protein named “mTOR” by autoradiography [19]. At the same time, Brown et al. used a fusion protein GFK (GST fused with FKBP12) prebound with two structural variants of rapamycin to pull down MG-63, Jurkat, or rat basophilic leukemia (RBL) cell lysates. Silver staining of GFK-bound mTOR was identified in each cell line and further verified in bovine brain tissues [20]. Independently, Sabers et al. used a GST-FKBP12-rapamycin affinity matrix to isolate putative mammalian targets of rapamycin (mTOR) from rat brain and murine T-lymphoma cell extracts [21]. Meanwhile, using a yeast-two-hybrid system composed of human FKBP12 and a mouse embryonic cDNA PCR library, Chiu et al. also identified mTOR as a target for FKBP12/rapamycin complex [22]. In early 2000s, presence of mTOR was also observed broadly in other organisms including fission yeast ([23], based on sequence homology search as homologs to budding yeast *TOR1* and *TOR2*), *C. elegans* ([24], also based on BLAST homologous search for PIK proteins), and Drosophila ([25], through a tissue-specific genetic screen for recessive mutations regulating compound eye development or [26] through a Drosophila cDNA library screening coupled RACE (rapid amplification of cDNA ends) analysis).

## 4. Identification and Characterization of mTOR Signaling Pathway Components

Identification of mTOR complex components were largely driven by the discovery that the detergent CHAPS but not Triton X-100 is more suitable to retain mTOR complex integrity. The hint for mTOR being present in a protein kinase complex came from observations that (1) two mTOR substrates showed distinct responses to immunoprecipitated mTOR-mediated phosphorylation in vitro washed with detergents—4E-BP1 phosphorylation was significantly reduced upon mTOR precipitates washed with 1% NP40 or 1% CHAPS [27], while S6K phosphorylation was enhanced by precipitated mTOR washed by either 1% NP40 or 1% Triton X-100 [28]; and (2) a 35 KD protein was copurified with mTOR [19]; and (3) a gel filtration chromatography experiment showed that yeast TOR1 or TOR2 migrated at ~2 MD molecular weight far larger than TOR itself [29]. A series of elegant biochemical approaches were used to purify and characterize mTOR complex and signaling components subsequently, in combination with genetic or other approaches (Figure 1). 

### 4.1. Biochemical Approaches to Identify mTOR Signaling Pathway Components

#### 4.1.1. Raptor (Regulatory Associated Protein of mTOR) Was Identified by TOR-IPs

Identification of Raptor was independently achieved by two research groups by distinctive purification procedures. Kim et al. utilized an anti-mTOR antibody to immunoprecipitate mTOR binding proteins derived from 3 × 10^6^ HEK293T cells radiolabeled with ^35^S-methionine followed by DSP (dithiobis (succinimidyl propionate)) cross-linking. Raptor was detected by autoradiography [30]. In another study, Hara et al. took a three-step purification procedure to find Raptor, including precipitating mTOR-containing proteins from 5 × 10^9^ HeLa cells by 30% (NH_4_)_2_SO_4_, followed by cation-exchange chromatography, and lastly, an mTOR antibody-aided immunoprecipitation of mTOR containing fractions [27]. In yeast, Raptor homolog Kog1 was identified through biochemical purification procedures as well. Specifically, 5 L of yeast cultured in YPD medium expressing either 2× Myc-tagged TOR1 or 3× HA-tagged TOR2 at endogenous levels were lysed in buffer containing 0.5% Tween 20, followed by ion-exchange chromatography and immunoprecipitations by either anti-Myc or anti-HA antibodies. A mock purification with non-tagged TOR was included as the negative control. Gel bands from purifications were silver stained, excised, and analyzed by MALDI-TOF mass spectrometry [29]. This led to identification of yeast TOR1 and TOR2 complex components including KOG1 (Raptor), LST8 (GβL), AVO1 (Sin1), AVO2, and AVO3 (Rictor). 

#### 4.1.2. GβL (G Protein β-Subunit-Like Protein) Was Identified by TOR-IPs

GβL was originally identified with Raptor in mTOR immunoprecipitants from HEK293T cells [30] and further confirmed by mTOR immunoprecipitations from 2 × 10^8^ HEK293T cells lysed with CHAPS buffer, using blocking peptides towards the mTOR antibody as a negative control [31]. In yeast, LST8 (GβL) was biochemically purified from yeast as described above [29]. 

#### 4.1.3. Sin1 (Stress Activated MAP Kinase-Interacting Protein, or MAPKAP1) Was Identified by Differential Raptor and Rictor-IPs

Given it has been determined that Raptor is unique to mTORC1 while Rictor depicts mTORC2, Jacinto et al. immunoprecipitated either Rictor or Raptor using a 0.3% CHAPS buffer, and coimmunoprecipitated proteins were analyzed by mass spectrometry. Sin1 was identified to be specifically associated with Rictor but not Raptor, which was further confirmed by proteomic analysis of the Sin1 immunoprecipitated complexes [32]. In a separate study, TAP-tagged GβL and TAP-tagged Raptor stably expressed in HEK293T cells were used as baits to pulldown 150–200 mg HEK293T whole cell extracts, and Sin1 was identified by mass spectrometry analysis as a protein binding to GβL but not to Raptor [33]. Further Sin1 depletion and deletion studies revealed that Sin1 maintains mTORC2 complex integrity and kinase activity [32,33]. Thus, Sin1 was identified as an essential component of mTORC2, but not of mTORC1. 

#### 4.1.4. PRAS40 Was Identified by Salt-Dependent Raptor/mTOR-IPs 

Identification of PRAS40 as an mTORC1-binding partner and activity suppressor was triggered by the observation that in mTORC1 in vitro kinase assays, mTORC1 complexes purified from either serum-starved or insulin-stimulated HEK293E cells, showed similar kinase activities towards phosphorylating S6K, whereas in cells, compared with serum starvation conditions, insulin strongly facilitated mTORC1 activation. Thus, Sancak et al. speculated that a negative mTORC1 suppressor might be lost during mTORC1 preps from HEK293E cells with high salt buffers being used [34]. By reducing salt concentrations to 150 mM NaCl from 300 or 400 mM NaCl, Sancak et al. observed an increased mTORC1 kinase activity in vitro isolated from insulin-stimulated cells [34]. Using Raptor or mTOR-IPs washed with either 150 mM NaCl or 400 mM NaCl, a ~40 KD protein PRAS40 was identified associated with mTOR and Raptor only under low salt concentrations. The PRAS40′s ability in suppressing mTORC1 kinase activity was further confirmed in both in vitro and in cell systems [34].

#### 4.1.5. DEPTOR Was Identified in Low Salt mTOR-IPs 

Using the same low salt buffer condition (150 mM NaCl, 1% Triton) for PRAS40 characterization [34], mTOR-IPs from HEK293E cells also led to identification of a 48 KD DEPTOR protein with an N-terminal DEP domain and a C-terminal PDZ domain [35]. Further DEPTOR-IPs confirmed that DEPTOR was associated with both mTORC1 and mTORC2 complexes, and depletion of DEPTOR in cells led to increased activities of both mTOR complexes [35]. These data support DEPTOR as an endogenous mTOR inhibitor.

#### 4.1.6. Proctor Was Identified as an mTORC2 Associating Protein Using Rictor-IPs

In an effort to identify novel mTORC2-binding proteins, endogenous Rictor IPs or TAP-tagged Rictor pulldowns were performed in either serum-starved or IGF-1-stimulated HEK293 cells [36]. A preimmune IgG antibody IP was included as a negative control. Protein bands present in Rictor-IPs but not IgG-IPs were excised for mass spectrometry analysis, which led to identification of Proctor-1 and Proctor-2. Further endogenous Proctor-1 or Flag-Proctor-1 IPs in HEK293 cells revealed that Proctor-1 associated with mTORC2 but not with mTORC1 complexes. Interestingly, Proctor-1 binding to mTOR or GβL was sensitive to certain detergents such as 1% Triton or 0.3% *n*-octylglucoside [36]. siRNA-mediated Proctor-1 knockdown did not affect mTORC2 complex association [36], but Proctor-1 might be necessary for mTORC2 to phosphorylate certain substrates such as SGK1 [37].

#### 4.1.7. TBC1D7 as the Third TSC Complex Component Was Identified by TSC2/TSC1 IPs

The TSC2/TSC1 functions as a heterodimer [38] to utilize its GAP activity to suppress mTORC1 activity. Flag-epitope tagged TSC2/TSC1 complexes were immunopurified by Flag antibody-conjugated beads from HEK293 cells with 1% NP40. Bound proteins were eluted with 3× FLAG peptides dissolved in nondetergent containing buffers and subjected to mass spectrometry analyses. The most abundant protein was TBC1D7 that associates with TSC1 to stabilize the TSC2/TSC1 complex towards suppressing mTORC1 [39].

#### 4.1.8. Rheb as a TSC2/TSC1 GAP Substrate Was Identified by the Biochemical Isolation of GTP/GDP Bound Rheb

Rheb was firstly characterized as a GTPase from screening of mRNAs induced in neurons by seizures-provoking agents [40]. In fission yeast, deletion of *Rhb1*, the yeast homolog of mammalian Rheb, mimicked nitrogen starvation to cause growth arrest [41]. To identify TSC2/TSC1 GAP substrates, Garami et al. utilized GST-fused GTPase activity probes composed of GTPase/GTP-binding domains [41] to pull down GTPases from serum-starved *TSC2^+/+^* and *TSC2^−/−^* MEF cells lysed with 1% NP40 [42], to search for increased GTPases activities upon TSC2 deletion. Due to low binding affinity of Rheb with TSC2/TSC1, this assay turned out to be unsuccessful. Instead, Rheb was immunoprecipitated from ^32^P-labeled, serum-starved *TSC2^+/+^* and *TSC2^−/−^* MEFs, respectively, and the ratio of GTP/GDP bound Rheb was determined by one-dimensional thin-layer chromatography (TLC). Using this approach, GTP-bound Rheb was enriched from *TSC2^−/−^* MEFs [42], suggesting Rheb is a candidate of TSC2/TSC1 GAP substrates.

#### 4.1.9. In Vitro GAP Activity Assays Identified Rheb as a TSC2/TSC1 GAP Substrate

Given that the TSC2 C-terminal region displays a sequence homology to RapGAP, Inoki et al. first purified GST-TSC2 C-terminal fusion proteins from *E. coli* and examined its GAP activity towards Ras subfamily GTPases, including Ras, Rap, TC21, and Rheb, and Rho family GTPases, including Rac and Cdc42 [43]. However, no obvious GAP activity was observed. Instead, Inoki et al. expressed TSC2 with TSC1 in HEK293 cells and immunoprecipitated TSC2/TSC1 complexes to perform in vitro GAP activity assays using the same set of small GTPases as substrates. As a result, TSC2/TSC1 was observed to stimulate GTP hydrolysis of Rheb in vitro [43].

#### 4.1.10. GST-Rheb Pulldown Identified Rheb as an mTORC1 Activator

The evidence for an association of Rheb with mTOR was obtained by using GST-Rheb proteins as baits to pull down endogenous mTOR. As a result, the amino-terminal portion of mTOR kinase domain (aa 2148–2300) was mapped as a Rheb-binding region [44]. The mechanism for how Rheb activates mTORC1 remained elusive until it was observed by confocal imaging that Rheb localized with mTOR on lysosomes upon growth factor stimulation for mTORC1 activation [45]. Notably, a weak endosomal membrane association of Rheb through its C-terminal farnesylation may also facilitate its association with mTOR for mTORC1 activation using a CSU-X1 spin disc confocal scan head [46] bearing a faster scanning speed and a higher use efficiency of the laser transmittance to significantly increase image resolution compared with canonical confocal microscopes.

#### 4.1.11. Rag GTPases Were Identified as mTORC1 Regulators Using a Modified Raptor-IP Protocol

Due to the presence of heavy and light chain signals in canonical antibody-based immunoprecipitants when resolved on SDS-PAGE, Sancak et al. speculated that some mTORC1 associating proteins might be masked by signals from antibody-derived heavy and light chains [47]. To this end, Flag-Raptor immunoprecipitants were obtained from 30 million HEK293T cells stably expressing Flag-Raptor and eluted with Flag peptides. Resulting proteins were resolved by SDS-PAGE, silver stained, and the 45–55 KD region (where the heavy chains locate) was excised and digested with trypsin for liquid chromatography and mass spectrometry analyses. Sancak et al. found a 44 KD protein RagC as a Raptor/mTORC1 associating protein. Further biochemical studies revealed that a heterodimer composed of GTP-loaded RagA/B and GDP-loaded RagC/D activated mTORC1 in responding to amino acids [47].

#### 4.1.12. Ragulator (MP1, p14, and p18) Was Identified as a Rag GTPase Binding partner by Rag-IPs to Recruit Rags to Lysosome

Considering that Rag GTPases localize to lysosome in an amino acid-independent manner [48], and Rag proteins lack the lipidation sequence that helps to recruit Rags to lysosome, Sancak et al. hypothesized that unknown protein(s) facilitates anchoring of Rag GTPases to lysosome. Mass spectrometry analyses of Flag-RagB or Flag-RagD immunoprecipitants from HEK293T cells led to identification of a trimeric protein complex composed of MP1, p14, and p18 that was termed as “Ragulator” as novel Rag GTPases binding partners. It was shown that the p18 N-terminal myristoylation and palmitoylation were critical for p18 localizing to lysosome membrane to recruit other Ragulator components and Rag GTPases to lysosome [48].

#### 4.1.13. HBXIP and C7orf59 Were Identified as Expanded Ragulator Components by p18/p14-IPs

Although Ragulator was identified as Rag GTPases binding partners in cells to recruit Rag GTPases to lysosome [48], in an in vitro cell-free system with purified recombinant proteins, Rag heterodimers interacted weakly with Ragulator [48]. This suggests that additional proteins might be needed to stabilize Ragulator/Rag interactions in cells. To identify such proteins, similarly Bar-Peled et al. performed Flag-p18 or Flag-p14 immunoprecipitations from HEK293T cells followed by mass spectrometry-mediated protein identification [49]. This led to identification of HBXIP and C7orf59 as expanded members of the Ragulator complex that exert a GEF activity towards RagA and RagB in promoting mTORC1 activation on lysosome in an amino acid- and v-ATPase-dependent manner [49]. Notably, in a recent study using quantitative methods to determine Ragulator-mediated binding affinity and kinetics of nucleotides to Rag GTPases (using bacterially purified Ragulator complexes mixed with purified Rag GTPases hetero-dimmers in a 1:1 ratio in vitro in the presence of radioactively labeled nucleotides) revealed that Ragulator also accelerates the release of GTP from RagC by opening the RagC nucleotide-binding pocket [50]. 

#### 4.1.14. The GATOR Complex as a GAP for Rag GTPases Was Identified through a Series of IPs

Upon identification of expanded Ragulator as a GEF for RagA and RagB that promotes mTORC1 activation, Bar-Peled et al. went on to examine how amino acid withdrawal inhibits mTORC1. Considering that the weak binding between GTPases including RagA and RagB with GTPases interacting proteins under conventional immunoprecipitation conditions, Bar-Peled et al. cross-linked Flag-RagB interacting proteins in HEK293T prior to Flag-immunoprecipitation and mass spectrometry-mediated protein identification. This led to identification of Mios as a new RagB interacting protein [51]. However, in vitro recombinant Mios proteins failed to bind Rag heterodimers, suggesting additional proteins are involved in this binding. Using Flag-Mios immunoprecipitants from HEK293T cells, additional seven proteins including WDR24, WDR59, Seh1L, Sec13, DEPDC5, Nprl2, and Nprl3 were found to be associated with Mios. Further Flag-DEPDC5 immunoprecipitants from HEK293T cells revealed a stronger Nprl2 and Nprl3 association with DEPDC5. Similarly, Flag-Npr12 immunoprecipitants from HEK293T cells demonstrated more abundant Npr12 binding to DEPDC5 and Npr13. Cumulatively, these series of immunoprecipitation experiments suggest the existence of two GATOR subcomplexes including GATOR1 (DEPDC5, Nprl2, and Nprl3) and GATOR2 (Mios, WDR24, WDR59, Seh1L, and Sec13) [51]. Further biochemical studies support GATOR1 as a GAP for RagA and RagB [51]. 

#### 4.1.15. FLCN/FNIP2 Were Identified as a GAP for RagC/RagD by Rag-IPs

Tsun et al. surprisingly found that GDP-loaded RagC was important for mTORC1 association with Rag heterodimers for mTORC1 activation [52]. To understand the regulatory mechanisms for RagC GTP/GDP loading, Tsun et al. performed Flag-RagA, RagB, RagC, or RagD immunoprecipitations-coupled mass spectrometry analyses in HEK293T cells and found FLCN and FLCN-binding proteins FNIP1 and FNIP2 as common binding partners for all Rags. Further in vitro GAP activity assays revealed that purified recombinant FLCN-FNIP2 proteins were able to stimulate GTP hydrolysis on RagC/RagD, but not RagA/RagB, supporting a role of FLCN/FNIP2 as a GAP for RagC and RagD [52].

#### 4.1.16. Sestrins as GATOR2 Binding Partners Negatively Regulating mTORC1 Activity Was Determined by Flag-GATOR2-IPs

In an effort to understand how GATOR2 is regulated, Chantranupong et al. performed Flag-GATOR2 (Flag-WDR24, Flag-Mios or Flag-WDR59) immunoprecipitations from HEK293T cells and found from mass spectrometry analyses that Sestrin 2 (Sestrin 1 and 3 to a lower level) was a constant binding partner shared by all GATOR2 components [53]. Further biochemical assays indicated that Sestrin 2 preferred GATOR2 but not GATOR1 for binding in cells and that Sestrin 2 binding to GATOR2 did not affect GATOR2 interaction with GATOR1. Sestrins binding to GATOR2 restrained mTORC1 localization to lysosome; thus, Sestrins serves as negative regulators for mTORC1 [53]. 

#### 4.1.17. GATOR2 as a Sestrin 2 Downstream Target Was Identified through Serstin 2 Pulldowns

Parmigiani et al. also identified Sestrins as GATOR2 binding partners but through a different route [54]. It was previously reported that as a stress-responsive gene, Sestrin 2 inhibited mTORC1 signaling through the AMPK/TSC signaling axis [55,56]. However, when Sestrin 2 and S6K were cotransfected into immortalized *AMPK*α*^−/−^* MEFs, Sestrin 2 was able to suppress mTORC1-mediated S6K phosphorylation even in the absence of AMPKα. Further, Sestrin 2-mediated inhibition of S6K phosphorylation in HEK293T cells could be partially released by the AMPK inhibitor compound C. These data suggested that in addition to the Sestrin 2/AMPK/TSC signaling, there was another yet unknown parallel signaling playing a role in Sestrin 2-governed mTORC1 suppression. To identify additional Sestrin 2 downstream targets in inhibiting mTORC1, Parmigiani et al. performed a tandem affinity purification from human mammary epithelial MCF10A cells stably expressing SBP (Streptavidin-Binding Peptide)-Flag-Sestrin 2 and identified GATOR2 proteins including Mios, WDR59, and WDR24 as Sestrin 2 binding partners. In addition, Kim et al. identified GATOR2 as Sestrin 2 binding partners by a similar tandem purification approach using Sestrin 2 as the bait [57].

#### 4.1.18. SLC38A9 as a Lysosomal Arginine Sensor for mTORC1 Activation Was Identified by Flag-GATROR2 and RagB-IPs

Identification of v-ATPases as a required factor governing mTORC1 activation through an “inside-out” mechanism by sensing lysosomal amino acid concentrations [58] suggests the existence of possible amino acid sensor(s) on lysosome. In brief, 30 million of HEK293T cells stably expressing individual GATOR2 components or Rags, including Flag-tagged p18, p14, HBXIP, c7orf59, and RagB, were used for Flag-immunoprecipitations under nonheated conditions and eluted by Flag peptides (Flag-METAP2 was used as a negative control) [59]. A lysosome membrane amino acid transporter SLC38A9 was a common hit in mass spectrometry analyses and shown to play a role in sensing lysosomal arginine sufficiency in activating mTORC1 [59]. Interestingly, in a follow-up study, Wyant et al. found that essential amino acids were accumulated in lysosome in SLC38A9-deficient cells and unraveled an unexpected function of SLC38A9 in regulating amino acid homeostasis through transporting essential amino acids out of lysosome in an arginine-dependent manner [60]. Recently, the molecular mechanism for SLC38A9-mediated mTORC1 activation was revealed by identification of SLC38A9 as a GEF for RagA. Specifically, bacterially purified SLC38A9N (aa 1–119) proteins were observed comigrated with Rag GTPases in vitro in a gel filtration experiment, and a stoichiometric amount of SLC38A9N proteins were found to inhibit the incorporation of radioactively labeled nucleotides into RagA in vitro by increasing the GDP off-rate [50].

#### 4.1.19. SZT2 (KISTOR)-IPs Identified GATOR1/GATOR2 as Its Binding Partners

Given all GATOR complex components are evolutionarily conserved in yeast within a single SEA complex that contains more members than GATOR1 and GATOR2, as well as GATOR1 and GATOR2 are loosely linked, Peng et al. reasoned that additional component(s) might be present to connect GATOR1 to GATOR2. Peng et al. identified SZT2 as a Sestrin 2 binding protein, and mass spectrometry analyses of Flag-SZT2 immunoprecipitants from HEK293T cells identified GATOR1 and GATOR2 as SZT2 binding partners [61]. Interestingly, SZT2 deficiency weakened GATOR1 and GATOR2 interactions, thus SZT2 was proposed to glue GATOR1 and GATOR2 to form the SOG (SZT2-orchestrated GATOR) complex. Notably, this interaction was not sensitive and regulated by amino acids. SZT2 truncation mutants deficient in maintaining GATOR1/GATOR2 interactions lost their ability to regulate mTORC1 signaling, indicating an indispensable role for an intact SOG complex in SZT2-mediated mTORC1 control [61].

#### 4.1.20. KISTOR Was Identified as a DEPDC5 Binding Protein Using Endogenous Flag-DEPDC5-IPs

In an effort to search for GATOR1 binding proteins, Wolfson et al. used CRISPR to knock in a Flag tag in front of the *DEPDC5* gene, an essential GATOR1 complex component and performed Flag-immunoprecipitations coupled with mass spectrometry analyses in HEK293T cells [62]. Four proteins including KPTN, ITFG2, C12orf66, and SZT2 were determined as new DEPDC5 binding partners, and this protein complex was named as KISTOR. Further biochemical and mouse genetic experiments demonstrated a role for KISTOR in binding and recruiting GATOR1 to lysosome for mTORC1 suppression under nutrient-limited conditions [62]. 

### 4.2. Genetic Approaches in Identification of mTOR Signaling Components

In addition to the powerful biochemical protein purification approaches, key mTOR signaling components were also identified by genetic approaches in yeast, Drosophila, and mammalian cells as detailed below.

#### 4.2.1. Genetic Association of TSC with TOR Was Discovered in Drosophila

Drosophila with homozygous *Tsc1* deletion died after hatching into first-instar larva, which could be partially rescued by loss of one copy of the *dTOR* gene [63]. Observations that *Tsc1*, *dTOR,* and double mutant cells are 1.9, 0.26, and 0.25 times the size of wild-type cells established an epistatic relationship that TOR functions downstream or in parallel to TSC2/TSC1 signaling in controlling cell size [63]. Further, a specific interaction between ectopically expressed TSC2 and dTOR was observed in Drosophila S2 cells [63]. 

#### 4.2.2. dRheb Was Identified as a TSC2/TSC1 Target Upstream of mTORC1 in a siRNA Screen

Given that a high mutation rate of GAP (GTPase activating protein) domain of TSC2 was observed in TSC patients and TSC2 shares a homologous GAP domain with Rap1-GAP, Zhang et al. reasoned that a GTPase would be a direct target for TSC2 in regulating mTORC1 activation [64]. Moreover, this GTPase would exert an opposite function from TSC2/TSC1 to maintain S6K-pT389 signals. By performing a siRNA-mediated screen in Drosophila S2 cells to search for loss of which GTPase(s) reduced S6K-pT389 [64], Rheb was identified as the GTPase negatively regulated by TSC2/TSC1. The GAP activity of TSC2/TSC1 towards Rheb was further confirmed by both in vitro and in cell assays [64].

#### 4.2.3. dRheb Was Identified in Regulating Cell Growth through Regulating mTOR Signaling in Drosophila Using Two Complementary Loss- and Gain-of-Function Genetic Screens

A loss-of-function screen was performed using ethyl methanesulfonate (EMS) to induce mutations, and mosaic Drosophila animals with reduced head size were further analyzed with disruptions in *Rheb* alleles through an *ey-Flp* mosaic screen [65]. In addition, a gain-of-function screen was performed using insertions of EP elements to drive gene overexpression. In animals with increased eye sizes, an EP element in the *Rheb* locus (EP 50.084) was identified [66]. Further genetic studies showed that *Pten* or *Tsc1/Tsc2* deficiency counteracted the effects of Rheb overexpression in promoting cell growth, and biochemical analyses revealed loss of *Rheb* that resulted in reduced S6K kinase activity using larval extracts from WT- or *Rheb*-deficient Drosophila [66].

#### 4.2.4. dRag GTPases Were Identified as mTORC1 Regulators by a siRNA-Mediated Screen in Drosophila 

As GTPases play roles in most known biological function, Kim et al. hypothesized that GTPase(s) may also be involved in amino acid-induced mTORC1 activation. To search for such potential GTPase candidate(s), an RNAi screen targeting all 132 annotated Drosophila GTPases was performed in Drosophila S2 cells, using amino-acid-induced dS6K phosphorylation monitored by Western blotting as a readout. As a result, dRagA and dRagC were hits from this siRNA screen [67]. 

#### 4.2.5. The GTPase Rap1 Was Identified to Suppress mTORC1 Activation through a siRNA-Screen

Mutvei et al. revisited their previous siRNA-mediated screen for mTORC1 regulators using ICW (in-cell Western) [68]. Specifically, HeLa cells were transfected with siRNAs, fixed and stained with an anti-rpS6 phospho-S235/S236 antibody (to monitor mTORC1 activity) followed by incubation with an IRDye-800CW-conjugated (infrared-excitable fluorophores) goat antirabbit secondary antibody. Resulting signals were quantitated by the LI-COR Aerius Infrared Scanner and normalized to cell number determined by staining cells with Alexa-680 succinimidyl ester. Using this technique, Mutvei et al. found the GTPase Rap1A as a top hit as a negative mTORC1 regulator, which could localize to lysosome. Depletion of Rap1A in HEK293A cells by two independent siRNAs reduced mTORC1 activation [69]. More importantly, using confocal imaging, Mutvei et al. observed that upon amino acid limitation, Rap1 suppressed lysosome abundance and its peripheral distribution. Thus, Rap1 suppresses mTORC1 activation under low amino acid conditions through limiting lysosome surface [69]. This provides an explanation for the low mTORC1 activity under amino acid deprivation conditions.

#### 4.2.6. The v-ATPase Stood Out in siRNA Screening as a Lysosomal Protein Regulating mTORC1

Given that Rag GTPases-mediated mTORC1 activation in responding to amino acids largely occurs at lysosome membrane, Zoncu et al. hypothesized that lysosomal proteins might participate in regulating mTORC1 activation. Using dsRNAs to deplete a panel of genes with characterized roles in controlling lysosome biogenesis and function in Drosophila S2 cells, Zoncu et al. found that depletion of only a small set of genes including vhaAC39, vha16, vha100-1, and vha100-2, all of which are components of the vacuolar H^+^-ATPase (v-ATPase), suppressed mTORC1-mediated dS6K phosphorylation [58]. These observations were further confirmed in mammalian cells and v-ATPase was found to be needed for Rag GTPases-induced mTORC1 activation on lysosome [58].

#### 4.2.7. v-ATPase and Arf GTPase Were Identified to Activate mTORC1 in Response to Glutamine by a Genetic Approach

Jewell et al. generated RagA/B KO (knockout) MEFs and unexpectedly found that loss of *RagA/B* reduced (~30%), rather than abolished, mTORC1 activity in responding to amino acid stimulation [70]. Through stimulating RagA/B WT and KO MEFs with each of individual 20 standard amino acids, Jewell et al. observed that RagA/B was required for leucine- or arginine-triggered mTORC1 activation, whereas glutamine stimulation minimally relied on RagA/B. Genetic and biochemical experiments identified Arf-GTPase and v-ATPase as hits that are required for glutamine-induced mTORC1 translocation to lysosome, a process independent of Rag GTPases [70]. 

#### 4.2.8. Rab35 Was Identified as an Oncogenic GTPase Activating Akt by a Genetic Screen

To search for potential regulators for the PI3K/Akt signaling, Wheeler et al. used a collection of 7450 shRNAs targeting kinases or GTPases to perform a loss-of-function genetic screen in HeLa cells using an in-cell immunofluorescent assay to monitor Akt-pS473 signal changes [71]. Rab35 stood out from this screen as a hit necessary for Akt activation and subsequent genetic and biochemical assays indicated Rab35 works upstream of PDK1 and mTORC2 to mediate PI3K-controlled Akt activation [71], while the detailed molecular mechanism remains to be determined.

#### 4.2.9. Ribosome Association Was Identified to Control TORC2 Activation by a Reverse Suppressor Genetic Screen in Yeast 

To identify mTORC2 upstream activators, Zinzalla et al. performed a reverse suppressor screen in budding yeast based on the observation that a yeast YPK2-D239A mutant [72] could rescue cell lethality caused by *TORC2* defect [73,74]. Zinzalla et al. hypothesized that potential mTORC2 activators are also depended upon YPK2-D239A for survival. This screen led to isolation of 44 independent mutants including mutants in core TORC2 complex such as TOR, AVO1, and AVO3, thus validating the screen [72]. More importantly, a *NIP7* temperature-sensitive mutant was also identified as a hit. Further genetic and biochemical assays in yeast and mammalian cells demonstrated that NIP7 was necessary for mTORC2 activation upon growth factor stimulation by regulating ribosome biogenesis but not protein synthesis, and ribosome association with mTORC2 was critical for insulin-PI3K-induced mTORC2 activation both in vitro and in cells [72].

### 4.3. Immunofluorescence-Mediated Identification of mTOR Signaling Components

Association of certain mTOR signaling components with mTOR was discovered by a direct visualization of their colocalization with mTOR in cells using high-resolution confocal microscopy, which includes the discovery of lysosome as the mTORC1 activation site and LRS (leucyl-tRNA synthetase) as a cellular leucine sensor to promote mTORC1 activation. 

#### 4.3.1. Amino-Acid-Stimulated mTORC1 Activation on Lysosome was Determined by Immunofluorescence

In 2010, to better understand mTORC1 cellular localization upon amino acid stimulation, Yasemin et al. stained human cells with antibodies detecting endogenous mTOR, Raptor, or RagC together with distinct endomembrane markers (not presented in the paper). Among them, LAMP2, a characterized lysosomal marker [75], was observed to colocalize with mTOR and Raptor only with sufficient amino acids [48]. On the other hand, an association of endogenous RagC with LAMP2 was observed as relatively stable and not sensitive to and regulated by amino acids. These observations suggest that cellular amino acids trigger mTORC1 recruitment to lysosome where Rag GTPases usually reside, and amino acids also induce Rag GTPases interactions with Raptor for mTORC1 activation [48].

#### 4.3.2. LRS Was Identified as an Intracellular Leucine Sensor by Immunofluorescence

Leucyl-tRNA synthetase (LRS) ligates leucine to its cognate transfer RNA. In an effort to search for tRNA-independent function of LRS, Han et al. performed cell fractionation and immunofluorescence assays and found a portion of LRS localized to lysosome at basal levels [76]. More importantly, amino acids stimulated LRS enrichment on lysosome to colocalize with mTOR, suggesting a possible role of LRS in regulating mTOR. Further biochemical assays indicated that associations of LRS with mTOR (by mTOR immunoprecipitations in HEK293T cells) were only observed in leucine-stimulated but not leucine-deprived conditions [76]. Further, LRS pulldowns in HEK293T cells showed that LRS specifically interacted with RagD in the presence of leucine and acted as a RagD GAP. The essential role of LRS as a leucine sensor for mTORC1 activation was confirmed by two types of LRS mutants: (1) an LRS-F50AL52A mutant deficient in binding leucine was not able to activate mTORC1 and (2)a tRNA transferase activity-deficient LRS-K716AK719A mutant did not affect mTORC1 activation [76].

### 4.4. Hypothesis-Driven Studies in Identification of mTOR Signaling Components

#### 4.4.1. Connection of TSC2/TSC1 with mTOR Was through an Observation That Both TSC2/TSC1 and mTOR Control S6K Phosphorylation

TSC2 and TSC1 form a heterodimer to suppress cell proliferation. Although TSC2/TSC1 have been tightly associated with cell size and proliferation control in mice [77,78] and Drosophila [79,80], the connection of TSC2/TSC1 with mTOR signaling has remained elusive till 2002. Inoki et al. found that overexpression of TSC2/TSC1 in HEK293 cells decreased phosphorylation of mTORC1 substrate S6K but not the mTORC2 substrate Akt, whereas siRNA-mediated TSC2 depletion in HeLa cells increased phosphorylation of S6K but not Akt, supporting a role of TSC2 in suppressing mTORC1 activity in cells [81].

Furthermore, TSC2 and TSC1 overexpression suppressed 4E-BP1 and S6K1 phosphorylation in U2OS cells. S6K1 mutants resistant to rapamycin (constitutively active) could not be suppressed by this TSC2/TSC1 ectopic expression, suggesting that mTOR mediates the suppressive effects of TSC2/TSC1 on S6K phosphorylation [82].

In addition, *TSC2^−/−^* MEFs displayed dramatically increased S6K phosphorylation at basal levels, which could be reversed by re-expressing WT, but not two TSC disease-associated mutant forms of TSC2, suggesting that *TSC2* loss promotes mTORC1 activation [83]. However, how TSC2/TSC1 connects to mTORC1 remained elusive until Rheb filled in.

#### 4.4.2. Hypothesis-Driven Discovery of Rheb as a GTPase in Activating mTORC1 

Discovery of a critical role of TSC2/TSC1 as a GAP (GTPase activating protein) to mTORC1 suggests that TSC2/TSC1 might function through an unknown GTPase to regulate mTORC1 activity. Castro et al. thus examined binding of a Flag-tagged C-terminal truncated version of TSC2 with a panel of GST-tagged Ras proteins including Rheb, Ha-Ras, R-Ras, Rap1A, and RalA loaded with GppNHp (nonhydrolyzable GTP analog) in the presence of MgCl_2_ in HEK293T cell lysates, and found that only Rheb displayed a strong binding ability. Following experiments further demonstrated that TSC2 was a Rheb GAP in cells and Rheb expression efficiently triggered S6K and S6 phosphorylation in cells [84]. 

#### 4.4.3. Hypothesis-Driven Approach to Identify Rheb in Activating mTORC1

Tee et al. hypothesize that Rheb might be a TSC2 GAP substrate to regulate mTORC1 based on the known knowledge that (1) *Rheb* deletion in fission yeast led to G0/G1 arrest that mimics nutrient starvation [41], (2) *Rheb* deficiency in fission yeast increased arginine and lysine uptake [85], and (3) as a Ras-related GTPase, together with Raf1, Rheb was able to transform NIH3T3 cells [86]. As a test result, overexpression of Rheb in HEK293E cells activated S6K1 but not Akt and RSK1, and TSC2 inhibited Rheb activity in HEK293E cells [87]. 

#### 4.4.4. Hypothesis-Driven Identification of Sestrin 2 as a Leucin Sensor for mTORC1 Activation

In mammalian cells such as HEK293T, removal of either leucine or arginine inhibited mTORC1-mediated S6K phosphorylation, consistent with these two types of amino acids being the major mTORC1 activity regulators. Given that the pentameric GATOR2 complex is required for amino acid-induced mTORC1 activation [51] and Sestrins (Sestrin 1, 2 and 3) were previously identified as a GATOR2 binding partner that suppresses mTORC1 activation with unknown mechanism(s) [53], Wolfson et al. examined if Sestrins binding to GATOR2 was regulated by amino acids and found that only leucine, but not arginine, depletion in HEK293T cells led to increased Sestrin 2 binding to GATOR2 [88]. Further evidence supported that leucine specifically disrupted Sestrin 2 binding to GATOR2 in cells and in vitro, thus establishing a critical role of Sestrin 2 as a leucine sensor to control mTORC1 activation—upon leucin availability, leucin releases Sestrin 2 suppression on GATOR2, leading to mTORC1 activation [88]. Further structure work with a complex of Sestrin 2 binding to leucine revealed the critical binding features [89]. 

### 4.5. Protein Motif/Sequence Homology Search-Mediated Identification of mTOR Signaling Components

#### 4.5.1. Mammalian Rictor Was Cloned and Sequenced Based on a Sequence Homology to Yeast AVO3

Yeast Rictor (AVO3) was discovered by 3× HA-mTOR immunoprecipitations as described [29]. The mammalian homolog of AVO3 was searched in a mammalian genome database. As a result, mammalian *Rictor* gene was identified and firstly cloned from mouse C2C12 cDNA and verified by sequencing [90]. By raising a specific antibody able to detect this molecule, binding of Rictor to mTOR and GβL was confirmed by coimmunoprecipitation assays in HEK293 cells [90].

#### 4.5.2. Mammalian Sin1 Was Identified by a BLAST Search of the Yeast AVO1 Gene

AVO1 (Sin1) was firstly identified as a TORC2 complex component in yeast through 3× HA-TOR2 IPs. A BLAST search identified putative AVO1 homologs in Drosophila (dSin1) and human (Sin1) [91]. RNAi-mediated dSin1 depletion in Drosophila S2 cells reduced dAkt-S505 phosphorylation with no impact on dS6K phosphorylation. Similarly, siRNA-mediated Sin1 depletion in human HeLa cells confirmed a critical role of Sin1 in governing Akt-S473 phosphorylation [91]. These data support an essential function of Sin1 in governing mTORC2 kinase activity towards phosphorylation of Akt hydrophobic motif. 

#### 4.5.3. PI(3,4,5)P_3_ Was Identified to Bridge PI3K/mTORC2 Through a Protein Motif Search

It is well characterized that PI3K is one of the major mTORC2 upstream activator [92], while the molecular connection between PI3K and mTORC2 remains elusive. Given that a PH domain was denoted in the C-terminus of the mTORC2 essential component, Sin1, and PH domains have long been characterized as phospho-lipid binding domains, Liu et al. examined if the Sin1-PH domain recognized PI(3,4,5)P_3_ produced by PI3K to promote mTORC2 activation. To this end, Liu et al. found that (1) Sin1-PH domain could functionally replace Akt-PH domain for Akt activation, suggesting Sin-PH was a functional PH domain in recognizing phospho-lipids; (2) recombinant Sin1-PH domain proteins recognized PI(3,4,5)P_3_ in lipid overlay assays; (3) overlaying Akt-PH and Sin1-PH structures led to identification of key PI(3,4,5)P_3_ binding resides; and (4) mutating PI(3,4,5)P_3_ binding residues in Sin1-PH led to reduced localization of Sin1-PH on plasma membrane in cells by immunofluorescence, attenuated Sin1-PH binding to PI(3,4,5)P_3_, and subsequently decreased mTORC2/Akt signaling in cells [93].

#### 4.5.4. Motif-Search-Based Identification of SLC38A9 as a Lysosomal Cholesterol Sensor for mTORC1 Activation

Given that mTORC1 promotes lipid synthesis through the mTORC1/Lipin1/SREBP1 signaling [94], Castellano et al. examined if dietary lipids control mTORC1 activity. This was based on several observations such as: (1) depletion of cellular cholesterol reduced mTORC1 activity in HEK293T cells, whereas exogenous LDL addition activated mTORC1 signaling; (2) sterol depletion still reduced mTORC1 activity in *TSC2^−/−^* MEFs; (3) manipulation of cholesterol levels had minimal effects on cellular amino acid levels; and (4) depletion of cholesterol weakened Ragulator binding to Rags. Thus, Castellano et al. hypothesized that cholesterol activated mTORC1 in a Rag GTPases dependent, but PI3K/Akt independent, manner (given this regulation occurs in *TSC2^−/−^* MEFs where the PI3K/Akt signaling disconnects from mTORC1 regulation) [95]. To nail down cholesterol targets in controlling mTORC1, Castellano et al. screened for cholesterol-regulated motifs among all the mTORC1 scaffolding complexes, which led to identification of a putative Cholesterol Recognition Amino Acid Consensus (CRAC) motif in SLC38A9. Using an ultraviolet light (UV)-photoactivatable cholesterol analogue, 7-Azi-27-yne, Castellano et al. proved a direct interaction between cholesterol and SLC38A9 in cells, and further biochemical studies supported a role of SLC38A9 in mediating cholesterol-induced mTORC1 activation [95].

### 4.6. Bioinformatics Approaches-Mediated Identification of mTOR Signaling Components

#### 4.6.1. CASTOR1 and CASTOR2 Were Identified as Arginine Sensors by a Bioinformatic Approach

SLC38A9 was identified as a lysosomal arginine sensor that suppresses mTORC1 activation when arginine is not available [59]. The observation that mTORC1 could be activated by arginine in cells lacking SLC38A9 suggests the existence of additional arginine sensors [59]. Considering that GATOR2 serves as an antenna to receive inputs from amino acids, Chantranupong et al. searched the BioPlex database (a human protein-protein interaction database [96]) for GATOR2 binding proteins and found GATSL3 (CASTOR1) as a GATOR2 binding partner, while GATSL2 (CASTOR2) interacted with GATOR1 [97]. Further biochemical studies supported a role of CASTORs in sensing arginine by binding to a distinct region on GATOR2.

#### 4.6.2. SAMTOR Was Identified to Connect One Carbon Metabolism to mTORC1 Activity Control by a Bioinformatic Approach

Gu et al. searched for GATOR1 or KISTOR binding partners using the BioPlex database and found C7orf60 (SAMTOR) was a common binding partner for both GATOR1 (Depdc5, Nprl3, and Nprl2) and KICSTOR (Kaptin, ITFG2, C12orf66, and SZT2) [98]. Subsequent biochemical analyses indicated SAMTOR was a *S*-adenosylmethionine sensor that links methionine/SAM to mTORC1 activity control [98]. 

## 5. Discussion

In this review, we summarize major experimental approaches leading to the identification of key mTOR signaling pathway components, including biochemical protein purification coupled mass spectrometry analyses, genetic approaches (such as yeast forward and reverse genetics and siRNA-mediated screens in mammalian cells), microscope-mediated immunofluorescent approaches, hypothesis-driven studies, protein motif or sequence homology searches, and bioinformatic approaches. These techniques serve as tools to examine ideas from scientists and the vast majority of the new discoveries are based upon previous findings. It is fascinating to read these fabulous studies and follow the logic flows of these discoveries in the mTOR field. It is enjoyable to find that most of these discoveries could be validated by either mouse genetic models (see [99] for review) or by structural insights (e.g., mTORC1 structures [100,101,102,103,104] and mTORC2 structure [105,106,107,108]). Due to the dynamic and complex nature of mTOR signaling, especially when taking feedforward and feedback regulatory loops into consideration, it is becoming difficult to utilize a single technique to obtain a full-frame image of mTOR signaling. Systems biology approaches have been developed to evaluate mTOR signaling regulations and dynamics using distinct mathematical models built upon available experimental data (e.g., [109,110]). Although as a promising and emerging direction to solve complex biological questions, this approach is restricted by limited amount of experimental data available as inputs for the computational system. In addition, various experimental systems and conditions used in different studies further introduce errors and variables in mathematic models. Thus, conclusions from systems biological studies would need further experimental validations, and in this scenario, a combination of artificial intelligence-aided research directions and experimental designs coupled with traditional experimental validation may lead to improved understanding of the complexity of mTOR signaling. 

## Figures and Tables

**Figure 1 genes-11-00738-f001:**
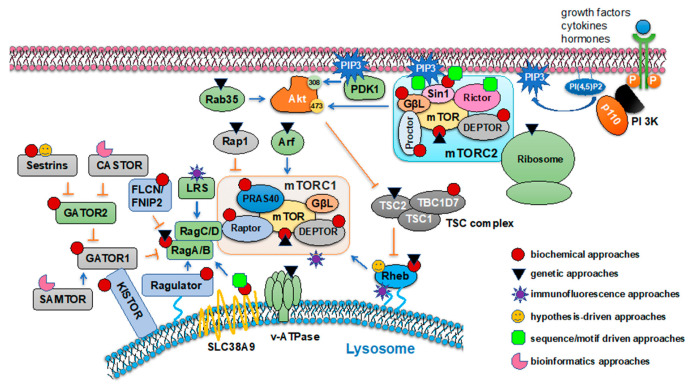
Summary of experimental approaches used in identification of major mTOR signaling components. This schematic illustrates six types of experimental approaches used in identification of key mTOR signaling pathway members by shapes denoted in the figure.

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
