# Peer review of "Experimental Approaches in Delineating mTOR Signaling"

_genes, 2020, doi:10.3390/genes11070738_

Round 1

Reviewer 1 Report

The review by Qian et al. focuses on experimental approaches for identification of factors in the mTOR signaling pathway. The authors comprehensively summarize how mTOR-related factors have been found. The manuscript is informative and interesting for the readership. However, prior to publication, grammar should be re-checked and some other corrections should be introduced:

  1. Line 58 and others. All of the following G-beta-L ‘beta’s are missing.

  1. Line 91. Allelic should be non-allelic.

  1. Line 93. The reference should be Heitman et al., Science, 1991.

  1. Lines 119 - 130 are difficult to understand. In line 120, the authors claim that CHAPS, but not Triton, is suitable for identification of mTOR components, while in line 124, they mention that CHAPS works negatively. All information is correctly quoted from the respective references, but the text should be reorganised.

  1. Figure 1. Some factor names, such as TSC1/TSC2, are missing.

  1. Line 193. Proctor1 should be Proctor2.

  1. Line 210. rhb1 is an essential gene in fission yeast. Deletion of Rhb1 causes growth arrest.

  1. Line 234. CSU-X1 is a product name of the spinning disc confocal unit. If necessary, the differences and benefits of this system compared to conventional methods should be explained.

  1. Line 238. It is hard to see where the modifications or improvements have been made compared to the previous approaches.

  1. Line 357. A more detailed explanation why Zhang et al. reasoned so would be easier to understand. Similarly, in Line 374.

  1. Line 380. In-cell western should be explained briefly.

  1. Line 421. It should be clarified whether it was demonstrated in yeast or mammalian cells.

  1. In lines 480 - 486, the main argument is unclear. The text should be reconsidered.

  1. Line 537. It is unclear in this context why cholesterol-mediated mTORC1 activation is independent of PI3K/Akt.

Reviewer 2 Report

This manuscript by Qian et al. summarizes the significant discoveries in the TOR signaling field from the perspective of the experimental approaches used to make these breakthroughs. It is a thorough historical summary of the complex TOR field that would help the lay reader, or an individual new to the TOR field, begin to understand the complex regulation of the pathway. The figure the authors have generated also is a well-done schematic that helps the reader visualize the complexity of the TOR pathway. The content of the article, and its organization, would be acceptable for publication in Genes. However, the article needs significant English language revision throughout the entire manuscript before being accepted for publication. This includes even the title since the “the” in the title makes it a poor title. Additional, specific points of concern are detailed below:

1) In the article, GβL is not properly presented- it shows up as G L. This needs to be corrected throughout the article.

2) Drosophila should always be capitalized. This needs to be corrected throughout the manuscript.

3) In Section 4.1.6, the authors refer to “Proctor1 and Proctor1”. This should be Protor-1 and Protor-2.

4) Line 213 should read “…MEF cells lysed…”, not “….MEF cells lyzed…”.

5) Line 236-27, “Sancak et al. speculated if any mTORC1….”, makes no sense.

6) In section 4.1.17, “Serstrin” should read Sestrin.

7) Line 334 mentions “SOG” without defining what it represents.

8) In section 4.4.3, line 482 the authors refer to budding yeast. This should be fission yeast.

9) In section 4.5.3, phosphor should be phospho.

Reviewer 3 Report

The authors review in the submitted manuscript different experimental approaches used in the mTOR signaling field from the TOR kinases discovery to the recent publications. This is an interesting overview of different techniques, which is of special interest to young investigators. I have only minor comments.

The authors announce at the beginning of the manuscript that they summarize major experimental approaches used in the past half a decade, but the vast majority of the cited literature is 10 to 30 years old.

Referring to a GEF activity of the Ragulator complex towards RAD A/B is formally correct, but this original finding was more recently revised by the authors (PMID: 30181260)

The statement (p.11) “overexpression of TSC2/TSC1 in HEK293 cells increased phosphorylation of mTORC1 substrate S6K…, TSC2 depletion in HeLa cells reduced phosphorylation of S6K…” is wrong.

Several corrections are needed to the Figure: Ragulator protein complex is lipid-anchored to the lysosomal membrane; SLC38A9 has 11 transmembrane regions, TSC complex is not labeled, the vATPase complex is hardly visible.

There are several typos and stylistic errors in the text, e.g., “a hierarch”, “phosphor-lipids”; mTOR senses … other microenvironment”; “mTORC1 responds to RAG/mTORC1…”; “approaches to identify mTOR components”, 014 should be p14.

Flag-METAP2 is not a component of GATOR2 as it is stated on P.8, but a negative control for co-IP experiments. Please correct.

The amount of starting material for pulldown experiments is presented as the total number of cells or mg of total protein.  The unification of the values to the total cell number would make different approaches more comparable.  

Round 2

Reviewer 2 Report

The section 4.1.6 still reads Proctor- this should be changed to Protor-1 or Protor-2.

Author Response

The section 4.1.6 still reads Proctor- this should be changed to Protor-1 or Protor-2.

Response: We thank the reviewer for raising this concern. In the revised manuscript, on lines 245 and 247 we specified Proctor-1 in the revised manuscript.